# Pepsinogen C Interacts with IQGAP1 to Inhibit the Metastasis of Gastric Cancer Cells by Suppressing Rho-GTPase Pathway

**DOI:** 10.3390/cancers16101796

**Published:** 2024-05-08

**Authors:** Hanxi Ding, Yingnan Liu, Xiaodong Lu, Aoran Liu, Qian Xu, Yuan Yuan

**Affiliations:** 1Tumor Etiology and Screening Department of Cancer Institute and General Surgery, The First Hospital of China Medical University, Shenyang 110001, China; dinghanxi0625@126.com (H.D.); liuyingnan_nnn@163.com (Y.L.); guardian9@163.com (X.L.); lareina0311@163.com (A.L.); 2Key Laboratory of Cancer Etiology and Prevention in Liaoning Education Department, The First Hospital of China Medical University, Shenyang 110001, China; 3Key Laboratory of GI Cancer Etiology and Prevention in Liaoning Province, The First Hospital of China Medical University, Shenyang 110001, China

**Keywords:** pepsinogen C, gastric cancer, biological behavior, tumor metastasis, IQGAP1

## Abstract

**Simple Summary:**

As the precursor of pepsin, Pepsinogen C (PGC) is the final product of the maturation and differentiation of gastric mucosa cells. The latest bioinformatics analysis found that PGC may play a little-known role in the carcinogenic process, but there is no relevant experimental evidence to prove the effect of PGC on the biological function of cancer cells. In our research, we found PGC may act as a tumor suppressor in the development and metastasis of gastric cancer. PGC can downregulate its interacting protein IQGAP1 and inhibit the Rho-GTPase pathway, thereby participating in the inhibition of gastric cancer cell migration and invasion. Thus, PGC has the potential to be a new therapeutic target for gastric cancer.

**Abstract:**

Aim: This study systematically explored the biological effects and mechanisms of PGC on gastric cancer (GC) cells in vitro and in vivo. Method: The critical biological roles of PGC in GC were assessed via EdU staining, Hoechst staining, flow cytometry, mouse models, CCK-8, wound healing, transwell, and sphere-forming assays. The interaction study with IQ-domain GTPase-activating protein 1 (IQGAP1) was used by Liquid chromatography-mass spectrometry co-immunoprecipitation, immunofluorescence staining, CHX-chase assay, MG132 assay, and qRT-PCR. Results: PGC inhibited the proliferation, viability, epithelial–mesenchymal transition, migration, invasion, and stemness of GC cells and promoted GC cell differentiation. PGC suppressed subcutaneous tumor growth and peritoneal dissemination in vivo. The interaction study found PGC inhibits GC cell migration and invasion by downregulating IQGAP1 protein and IQGAP1-mediated Rho-GTPase signaling suppression. In addition, PGC disrupts the stability of the IQGAP1 protein, promoting its degradation and significantly shortening its half-life. Moreover, the expression levels of PGC and IQGAP1 in GC tissues were significantly negatively correlated. Conclusion: PGC may act as a tumor suppressor in the development and metastasis of GC. PGC can downregulate its interacting protein IQGAP1 and inhibit the Rho-GTPase pathway, thereby participating in the inhibition of GC cell migration and invasion.

## 1. Introduction

The *pepsinogen C* (*PGC*) gene is located on chromosome 6p21.1. It contains 11 exons and encodes a 42-kDa protein [1]. As a member of the aspartic protease family, PGC protein is synthesized by the chief cells of the gastric mucosa and secreted into the gastric lumen. PGC is activated to active pepsin C under the acidic conditions of the stomach, and thereby participates in the digestion of proteins in the stomach [2,3]. PGC is mainly expressed in the whole stomach as the final product of gastric mucosal cell maturation and differentiation under normal physiological conditions. However, in the dynamic cascade process from normal gastric mucosa to superficial gastritis, chronic atrophic gastritis, intestinal metaplasia, dysplasia, and finally to gastric cancer (GC), PGC expression level is successively decreased [4,5,6]. In GC, PGC expression in situ is almost completely absent, indicating that PGC is a highly distinct negative tissue marker for GC [2,7]. Because PGC can be secreted into the blood, its expression levels can be determined by serology. The role of serum PGC and its isoenzyme pepsinogen A (PGA) as early warning markers of gastric precancerous lesions and GC has been widely recognized, and some research results have been applied in clinical practice [8,9,10,11].

Although PGC has long been considered to be the precursor of pepsinogen and the final product of the maturation and differentiation of gastric mucosal cells, recent studies have found that PGC may have other, lesser-known functions beyond those mentioned above. A pan-cancer analysis of the pepsinogen gene family based on the Cancer Genome Atlas (TCGA) database revealed that PGC is associated with the activation or inhibition of many signal transduction pathways, especially tumor pathways, which are mainly involved in GC, esophageal cancer, and lung squamous cell carcinoma-related regulatory network pathways. These findings suggested that PGC may be a regulatory factor that plays a crucial role in tumorigenesis and progression [12]. At present, there is no relevant experimental evidence of the effect of PGC on the biological functions of cancer cells. A comprehensive and in-depth exploration and evaluation of the effects of PGC on cancer cells’ biological functions and the underlying molecular mechanisms will help to explain the role of PGC in the occurrence and development of cancer. It will also provide a new theoretical basis for the prevention, diagnosis, and treatment of cancer and deliver a potential therapeutic target.

GC is one of the most common gastrointestinal malignancies in the world, and its morbidity and mortality rank fifth and fourth, respectively [13]. In this study, we systematically explored the effects of PGC on the biological functions of GC cells from the in vitro and in vivo levels. At the same time, we attempted to identify PGC-interacting proteins and whether their interactions inhibit the migration and invasion of GC cells. In addition, we examined the effects of a human-derived PGC intervention on the growth and intraperitoneal dissemination of tumors in nude mice. Our aim was to elucidate whether PGC plays a regulatory role in GC and has the potential to be a new therapeutic target for GC in addition to its known role as an early warning marker for GC and its precancerous diseases.

## 2. Materials and Methods

### 2.1. Cell Culture and Lentivirus Infection

HGC-27, AGS, and MKN-45 cell lines were purchased from the Institute of Basic Medical Science Chinese Academy of Medical Sciences (Beijing, China). PGC-FLAG-EGFP lentiviral vector and the negative control vector were purchased from Shanghai Genechem Company (Shanghai, China), and cells were transfected according to the manufacturer’s instructions. Stably transduced cell lines were obtained by selection with Puromycin for PGC and Neomycin for IQ-domain GTPase-activating protein 1 (IQGAP1). All of the details of the experimental materials are shown in the Appendix A.

### 2.2. RNA Extraction and Quantitative Real-Time PCR

RNA extraction and quantitative real-time PCR were performed (as shown in the Appendix A) and β-actin was used as an internal reference for calibration. All primer sequences used in this study are shown in Appendix A.

### 2.3. Protein Extraction and Western Blot Assay

Protein extraction and Western blot assay are described in detail in the Appendix A. The primary antibodies used in this study are as follows: anti-PGC (1:1000; #ab255826, Abcam, Cambridge, UK), anti-IQGAP1 (1:1000; #ab133490, Abcam), anti-ARHGEF2 (1:2000, #ab201687, Abcam), anti-CDC42BPA (1:1000, #ab146566, Abcam), anti-ENO1 (1:1000, #227978, Abcam), anti-Rho (1:1000, #ab188103, Abcam), anti-FLAG (1:1000, AF519, Beyotime Biotechnology, Shanghai, China), and anti-HA (1:1000, AH158, Beyotime Biotechnology). β-tubulin (1:1000, abs830032, Absin, Shanghai, China) and GAPDH (1:1000, abs830030, Absin) were used as loading controls.

### 2.4. Enzyme-Linked Immunosorbent Assay

The PGC expression level in the cell culture supernatant was determined by using a pepsinogen II enzyme-linked immunosorbent assay (ELISA) kit as per the instructions (BioHit, Helsinki, Finland), and the details of the process are described in the Appendix A.

### 2.5. 5-Ethynyl-2′-deoxyuridine Incorporation Assay

GC cells were seeded in 24-well plates for the determination of DNA synthesis and cell proliferation using a 5-Ethynyl-2′-deoxyuridine (EdU) assay kit (Beyotime Biotechnology). The experimental process is described in the Appendix A.

### 2.6. Cell Counting Kit-8 Assay

GC cells were seeded in 96-well plates (1 × 10^4^ per well) and observed for 72 h, and cell viability was tested using a Cell Counting Kit-8 (CCK-8) solution (10 µL/well) every 24 h (MCE, Shanghai, China). After incubation at 37 °C for 1.5 h, the absorbance value at 450 nm was measured by microplate reader.

### 2.7. Cell Apoptosis Assay

GC cells were digested with EDTA-free trypsin and adjusted to 5 × 10^5^ per sample in 500 µL binding buffer and then incubated at room temperature for 15 min after the addition of 5 µL Annexin V-APC and 5 µL 7-AAD (KeyGEN BioTECH, Nanjing, China) in the dark. Finally, cell apoptosis was determined via a flow cytometer (BD, San Diego, CA, USA) and the apoptosis rate was calculated using FlowJo software version 10.

### 2.8. Hoechst Staining Assay

Seeded GC cells were stained with 10× Hoechst 33342 (Beyotime Biotechnology). The apoptosis rate was evaluated using a fluorescence microscope (Olympus, Tokyo, Japan). 

### 2.9. Cell Cycle Assay

For this assay, cells were stained with PI/RNase Staining Buffer (Thermo Fisher, Waltham, MA, USA) for 30 min in the dark and analyzed using flow cytometry (BD, USA). The cell cycle phase distribution was analyzed by ModFit software version 3.3.

### 2.10. Sphere Formation Assay

GC cells were inoculated in the ultra-low adhesion 24-well plates (Corning, Corning, NY, USA) at a density of 3000 cells per well and incubated with RPMI 1640 medium containing 2% B27 (Thermo Fisher), 20 ng/mL bFGF (Solarbio, Beijing, China), and 20 ng/mL EGF (MCE) for 7 days. To calculate the sphere formation efficiency, spheroids between 40 and 100 µm in size were counted using an inverted microscope (Olympus, Tokyo, Japan). 

### 2.11. Transmission Electron Microscopy (TEM) Assay

GC cell samples were fixed in 2.5% glutaraldehyde at 4 °C for 24 h. Subsequently, these samples were immobilized with 1% osmic acid, dehydrated with ethanol and acetone, and embedded in 812 embedding agent. Finally, the sectioned samples were stained with lead citrate and then observed by transmission electron microscopy (TEM) (HITACHI, Tokyo, Japan) to obtain representative images.

### 2.12. Wound Healing Assay

GC cells were seeded and cultured in 6-well plates until confluence, and then the cell layer was scratched with a 100 µL sterile pipette tip. After being scratched (recorded as 0 h), the cells were incubated in serum-free medium for 24 h (recorded as 24 h). The cell migration distance was captured and measured using a universal microscope (Olympus, Japan) and Image J (https://ij.imjoy.io/ (accessed on 23 April 2024)).

### 2.13. Transwell Assay

To assess cell migration, 3 × 10^4^ GC cells suspended in serum-free medium were seeded in the upper chamber, while RPMI 1640 with 10% FBS was added into the lower chamber. In the invasion assay, Matrigel (BD) was precoated on the upper membrane surface of a transwell chamber and then 5 × 10^4^ GC cells were inoculated in the chamber. The detailed information is shown in the Appendix A.

### 2.14. Co-Immunoprecipitation and Liquid Chromatography-Mass Spectrometry

Co-immunoprecipitation (Co-IP) assay was performed using an immunoprecipitation kit following the manufacturer’s guidelines (Beyotime Biotechnology), and the details are described in the Appendix A.

### 2.15. Co-IP of PGC and IQGAP1 in GC Cells

According to the above method, the total protein of AGS cells that stably transfected with PGC (FLAG-tagged) overexpression vector or IQGAP1 (HA-tagged) overexpression vector was extracted. Lysate (at least 1 mg protein), and magnetic beads fused with the FLAG label or the HA label were incubated overnight at 4 °C. WB was used to determine the expression of PGC and IQGAP1 protein.

### 2.16. Cycloheximide and MG132 Assays

Cycloheximide (CHX) chase assay was performed to determine the half-life of the IQGAP1 protein. GC cells stably transfected with PGC overexpression vector or empty vector were treated with the protein synthesis inhibitor CHX (100 µM/mL) for 2 h, 4 h, 6 h, and 8 h. GC cells overexpressing PGC protein and the corresponding control cells were treated with the proteasome inhibitor MG132 (20 µM/mL) for 6 h to block the proteasomal degradation of IQGAP1 protein prior to cell lysing with lysis buffer. Then, the total protein of each time point was extracted and the expression level of the IQGAP1 protein was determined by WB.

### 2.17. Immunofluorescence Staining

GC cells were fixed with 4% paraformaldehyde and treated with Triton X-100. Cells were blocked with donkey serum at 37 °C for 30 min and then incubated with IQGAP1 antibodies (1:100; ab86064, Abcam) overnight at 4 °C. The next day, cells were incubated with a fluorescent secondary antibody (Thermo Fisher) and then stained with DAPI. Finally, protein fluorescence pictures were captured and evaluated using a fluorescence microscope.

### 2.18. In Vivo Animal Experiments

Specific pathogen-free grade male BALA/C nude mice, weighing 16–18 g and aged 4 to 5 weeks, were purchased from SPF Biotechnology Company (Beijing, China). All in vivo animal experimental protocols were approved by the Animal Care and Use Committee of the First Hospital of China Medical University (KT2022219). 

The xenograft subcutaneous injection was performed for the subcutaneous tumor formation experiment, and a peritoneal dissemination assay was performed via intraperitoneal injection, all using the MKN-45 cell line (see the Appendix A).

### 2.19. Bioinformatics Analysis

The differential expression levels of *PGC* and *IQGAP1* mRNA, as well as the expression correlation in 408 GC and 211 normal control tissues, were analyzed using the GEPIA2 database (http://gepia2.cancer-pku.cn/, accessed on 23 April 2024) and GTEx databases (https://commonfund.nih.gov/gtex, accessed on 23 April 2024) [14]. The correlation between *PGC*, the *IQGAP1* mRNA expression level, and GC-patient survival was analyzed via the Kaplan-Meier Plotter database [15]. In addition, we also analyzed the PGC and IQGAP1 protein expression differences in 80 GC and paired normal tissues using the CPTAC database (https://cptac-data-portal.georgetown.edu/, accessed on 23 April 2024) [16,17].

### 2.20. Patients and Tissue Specimens

Twenty paired pathologically confirmed GC and adjacent normal tissues collected from the First Hospital of China Medical University were used to evaluate the *PGC* and *IQGAP1* mRNA expression differences and the expression correlation in GC patients. All subjects gave their informed consent for inclusion before they participated in the study. The study was conducted in accordance with the Declaration of Helsinki, and the protocol was approved by First Hospital of China Medical University Ethics Committee ([2022]259).

### 2.21. Statistical Analysis

In this study, statistical analyses were carried out by SPSS 23.0 software and GraphPad Prism 5. Results are presented as means ± standard error of the mean (SEM). The two-tailed Student’s *t*-test was used for normally distributed data, while the rank-sum test was applied for skewed distributed data. A *p*-value < 0.05 was considered statistically significant.

## 3. Results

### 3.1. PGC Inhibits GC Cells Proliferation, Migration, Invasion, and Epithelial–Mesenchymal Transition In Vitro

PGC overexpression lentivirus (LV-PGC) and negative control lentivirus (LV-Ctrl) were transduced into HGC-27, AGS, and MKN-45 cells to explore the effects of PGC on the biological behavior of GC cells. In the CCK-8 assay, PGC overexpression significantly inhibited the viability of HGC-27, AGS, and MKN-45 cells (Figure 1A–C). In the EdU assay, PGC overexpression significantly suppressed the proliferative capacity of HGC-27 and AGS cells (Figure 1D–G). In the wound healing and transwell assays, PGC overexpression significantly reduced the migration and invasion capabilities of HGC-27 and AGS cells (Figure 1H–K). In the epithelial–mesenchymal transition (EMT) assay, *PGC* overexpression upregulated the epithelial marker (*E-cadherin*) and downregulated mesenchymal markers (*N-cadherin*, *vimentin*, *fibronectin*), metalloproteinases (*MMP2* and *MMP9*), and associated transcription factors (*Snail*, *Slug*, and *Twist*) (Figure 1L–N), suggesting *PGC* was involved in EMT of GC cells. The background expression of PGC in the three GC cell lines is shown in Appendix A. The overexpression status of PGC mRNA and protein is shown in Appendix A.

### 3.2. PGC Inhibits GC Cell Growth and Metastasis In Vivo

To confirm the effects of PGC on tumorigenicity and xenograft tumor growth in vivo, we established a subcutaneous xenograft model in nude mice. Appendix A shows the expression of PGC in the xenograft tissues. As shown in Figure 2A,B, the weight and volume of tumors of PGC-overexpressing GC cells was significantly smaller than that of the control group. To further clarify the inhibitory effect of PGC protein on xenograft tumor growth, we established a subcutaneous tumor model of nude mice using wild-type MKN-45 cells. The tumor-bearing mice were randomly divided into a human PGC active protein (hPGCp) injection group and an NS injection group. The results demonstrated that hPGCp significantly suppressed tumor growth compared with controls, as exhibited by the tumor weight and volume data (Figure 2C,D).

Because peritoneal metastasis is a frequent metastasis pattern in GC patients, peritoneal dissemination was assessed via the intraperitoneal injection of tumor cells. Compared with LV-Ctrl cells, PGC-overexpressing MKN-45 cells produced fewer peritoneal metastasis nodules and had a lower organ invasion rate. The number and weight of metastasis nodules in the LV-PGC group were lower than those in the LV-Ctrl group (Figure 2E,F). The invasion rates of the liver and pancreas were 60% and 100%, respectively, in the mice of the LV-Ctrl group, but they were 20% and 20%, respectively, in the mice of the LV-PGC group (Figure 2G). The H&E staining of all tumors from the mice confirmed that the pathological characteristics were consistent with malignant disease, and this was approved by a pathologist (Figure 2H). 

### 3.3. PGC Promotes Differentiation and Inhibits the Stemness of GC Cells

We examined the expression changes of mature differentiation markers and dedifferentiation markers in gastric mucosa epithelium in order to determine the potential effect of PGC on GC cell differentiation. Compared with control cells, PGC-overexpressing cells harbored higher expression levels of mature differentiation markers (*ATP4A*, *ATP4B*, *MIST1*, *NEUROG3*, *GAST*, *TFF1*, *TFF3*, *MUC5AC*, *MUC6*, and *GKNI*) and significantly lower expression levels of gastric epithelial stem cell marker (*VIL-1*), gastric epithelial progenitor cell markers (*SOX2* and *SOX9*), GC stem cell markers (*CD44* and *CD133*), and pluripotency genes (*OCT-4* and *Bmi-1*) (Figure 3A). The detailed results can be found in Appendix A. The results of qRT-PCR assays showed good evidence that *PGC* regulated the differentiation and stemness of HGC-27, AGS, and MKN-45 cells at the molecular level. At the same time, we also explored the effect of PGC on the stem-cell-like phenotype using sphere formation assays in GC cells. The results showed a significantly lower sphere-forming efficiency, with a smaller number and a lower sphere diameter size, in PGC-overexpressing HGC-27 and AGS cells (Figure 3B–E).

In addition, the subcellular morphological structure characteristics of GC cells were observed under TEM. The results suggested that, compared with LV-Ctrl cells, PGC-overexpressing HGC-27 and AGS cells showed more differentiated subcellular structure characteristics, with fewer cell surface microvilli, rough endoplasmic reticulum, and higher numbers of Golgi complexes and lysosomes with a high electron density, as well as clearer mitochondrial cristae and a shallower nuclear envelope depression (Figure 3F,G).

### 3.4. PGC Is Not Correlated with GC Cells Apoptosis or Cell Cycle Progression

We also analyzed the effects of PGC on GC cell apoptosis and cell cycle progression. The results of Hoechst staining of HGC-27 and AGS cells (Appendix A) and flow cytometry of HGC-27, AGS, and MKN-45 cells (Appendix A) showed that PGC did not significantly affect GC cell apoptosis and the phase distribution of cells cycle (Appendix A). 

### 3.5. PGC Interacts with IQGAP1 and Interferes with IQGAP1 Protein Stability in GC Cells

To explore the potential mechanism, we used FLAG fusion magnetic beads to conduct an immunoprecipitation assay and precipitated the PGC target proteins in GC cells. In total, 377 proteins were identified from the precipitated proteins by LC/MS analysis (Appendix A). According to the results of GO and KEGG analysis, these proteins were mainly located at the cell membrane and cell junction, and they were mainly related to cell localization at the cell-connection-associated pathways (Appendix A). Through annotation using the UniProt database, thirty-two proteins were found to be closely correlated to cell migration and invasion. Of these proteins, four (ARHGEF2, CDC42BPA, ENO1, and IQGAP1) were associated with the Rho-GTPase network, which plays a decisive role in cytoskeletal structure (Appendix A). Co-IP results revealed that PGC interacted with IQGAP1, but not ARHGEF2, CDC42BP2, or ENO1 (Figure 4A–C). Furthermore, FLAG-PGC and HA-IQGAP1 were cotransfected into AGS cells to perform a Co-IP assay. We found that HA-IQGAP1 immunoprecipitated with FLAG-PGC using an anti-FLAG antibody, suggesting that PGC can form a protein–protein complex with IQGAP1 (Figure 4D). In addition, immunofluorescence staining results demonstrated that PGC and IQGAP1 were colocalized in the cytoplasm of AGS cells (Figure 4E).

Additionally, results of WB and qRT-PCR assays demonstrated that PGC overexpression decreased IQGAP1 protein expression level rather than the mRNA levels in HGC-27, AGS, and MKN-45 cells (Figure 4F–H). Next, we attempted to elucidate the underlying molecular mechanism by which PGC downregulates IQGAP1 expression. We assessed the effect of PGC on the stability of the IQGAP1 protein using the CHX chase assay. WB results showed that the half-life of the IQGAP1 protein was advanced in PGC overexpression cells, suggesting that PGC decreased the protein stability of the IQGAP1 protein (Figure 4I). Furthermore, PGC-overexpressing cells and negative control cells were treated with MG132 for 6 h, and then the IQGAP1 protein expression level was determined using WB. The results indicated that the downregulation of IQGAP1 protein induced by PGC could be rescued by the proteasome inhibitor MG132 (Figure 4J). 

### 3.6. PGC Inhibits GC Cell Migration and Invasion by Reducing IQGAP1 and Suppressing Rho-GTPase Signaling

Because it has been proven that IQGAP1 is involved in cell migration and invasion and is a key node in the Rho-GTPase network, we hypothesized that PGC may exert its antioncogene effect by suppressing IQGAP1 expression and the Rho-GTPase pathway. To corroborate this speculation, we constructed PGC-overexpressing (LV-PGC + IQGAP1-NC), IQGAP1-overexpressing (PGC-NC + LV-IQGAP1), and the cotransfected (LV-PGC + LV-IQGAP1) GC cells for functional rescue tests. In HGC-27 and AGS cells, wound healing, cell migration, and invasion ability were significantly reduced in LV-PGC + IQGAP1-NC group cells and increased in LV-IQGAP1 + PGC-NC group cells, whereas increased expression of IQGAP1 counteracted the effects of PGC overexpression (Figure 5A–F). Subsequent Co-IP assay and WB revealed that Rho proteins interacted with IQGAP1 and the inhibitory effect of PGC on Rho protein expression (Figure 4G and Figure 5G).

### 3.7. PGC Expression in GC Is Significantly Negatively Correlated with IQGAP1

We initially compared *PGC* and *IQGAP1* mRNA expression levels in 408 GC and the corresponding 211 normal tissues using the GEPIA2 database. The results suggested that *PGC* was significantly downregulated while *IQGAP1* was highly expressed in GC tissues, and there was a significant negative correlation (R = −0.52, *p* < 0.001, Figure 6A–C). In addition, analysis results from the Kaplan-Meier Plotter database showed that low expression of *PGC* was related to poor overall survival in GC patients (Figure 6D,E). Along those lines, the analysis results of the CPTAC database showed that PGC protein expression level was also negatively correlated to IQGAP1 protein (R = −0.305, *p* = 0.006, Figure 6F–H). On this basis, we further determined the expression level of *PGC* and *IQGAP1* in 20 pairs of GC and matched adjacent normal tissues. Consistent with the database results, *IQGAP1* was highly expressed in GC tissues while PGC showed a lower pattern of expression in GC (Figure 6I). 

## 4. Discussion

PGC has long been considered a digestive enzyme precursor in biological processes [1,18]. Several investigations have validated the potential of serum PGC expression levels as indicators in screening for GC and its precancerous conditions, holding significant promise for clinical applications [8,9,11]. More recent bioinformatics analyses have found that PGC is related to the activation or inhibition of GC-associated signal transduction pathways, which strongly suggests that the functions of PGC might not be limited to digestive enzyme precursors and that it is likely to act as a regulatory factor in the development and metastasis process of GC [2]. However, this speculation is not yet supported by relevant experimental evidence. In this study, the effects of PGC on the biological behavior of GC cells and its mechanism were systematically investigated in vivo and in vitro. To our knowledge, this is the first study to report the biological functions of PGC outside of its original role as a digestive enzyme precursor. 

Identification of whether a gene promotes or inhibits the behavior of cancer cells can indicate whether the gene plays an oncogenic or tumor suppressor role [19,20,21,22,23]. This study explored the effects of PGC on various biological behaviors of GC cells to clarify the possible roles of PGC in GC development, invasion, and metastasis. The results of this study showed that PGC was significantly negatively correlated to GC cell proliferation and viability and had no significant effect on apoptosis and the cell cycle phase distribution. This indicates that the effect of PGC on the proliferation capacity of GC cells may either result in a reduced number of cells entering the cell cycle or a general slowdown of the cell cycle. Although this study did not delve into the effects on the cell cycle extensively, it presents a valuable direction for future research. In addition, it significantly inhibited the subcutaneous tumorigenic ability and peritoneal dissemination ability of GC cells in nude mice and upregulated multiple differentiation-related indicators of GC cells. The differentiation-like subcellular structure of GC cells after PGC overexpression was more obvious under electron microscope observation. PGC inhibited the expression of stemness-related markers of gastric mucosal stem/progenitor cells and significantly inhibited GC cell spheroidization ability. Furthermore, PGC inhibited the migratory and invasive ability of GC cells and downregulated mesenchymal cell markers (N-cadherin, vimentin, and fibronectin), metalloproteinases (MMP-2, MMP-9), and other metastasis-related proteins (Snail, Slug, Twist), and it upregulated epithelial cell marker (E-cadherin) expression, thereby inhibiting the EMT progression of GC cells. In addition, we constructed a simulated model of GC cell intraperitoneal dissemination and metastasis, which is the most common and aggressive metastatic mode of gastric cancer, and observed the effect of PGC on the dissemination ability of GC cells. The experimental results confirmed that PGC significantly inhibits the number and weight of peritoneal implanted tumors and the invasion of GC cells into peritoneal organs. To summarize, through a series of experiments in vitro and in vivo, we confirmed that PGC is negatively associated with the malignant biological behaviors of GC, such as proliferation, viability, migration, invasion, EMT, and stemness, and it is positively correlated with the maturation and differentiation of GC cells. Therefore, our evidence suggests that PGC plays a role as a tumor suppressor gene in the process of GC development, invasion, and metastasis.

On the basis of clarifying the biological functions of PGC, we further identified the PGC-interacting proteins and explored the mechanisms by which the GC cell migration and invasion were inhibited. In total, 377 proteins interacting with PGC were identified by Co-IP, WB, and mass spectrometry, which were primarily involved in metabolism-related pathways, protein-process-related pathways, the PI3K-AKT signaling pathway, and cell-connection-related pathways according to a bioinformatics analysis. Among the 377 proteins interacting with PGC, 32 proteins were closely related to cell movement, cell migration, cell adhesion, and other functions. Among these 32 proteins, we identified 9 proteins that may overlap with PGC in subcellular localization, including ARHGEF2, CDC42BPA, ENO1, IQGAP1, EPHA2, GPI, MYH9, S100A14, and YBX1. Among them, ARHGEF2, CDC42BP2, ENO1, and IQGAP1 are closely associated with the Rho-GTPase network and play pivotal roles in cytoskeletal organization [24]. Furthermore, through two-way Co-IP and immunofluorescence experiments, we found that PGC interacts only with IQGAP1 among these four proteins and co-localizes with it. The results suggested that IQGAP1 is an important interacting protein in the biological process of PGC.

IQGAP1 is a scaffold protein containing several domains that mediate protein interactions [25,26,27]. As an oncogene, IQGAP1 can bind to several proteins and then participate in various biological activities, such as cytoskeleton dynamics, cell adhesion, cell motility, cell invasion and metastasis, and cell proliferation [28,29,30]. Results of this study suggested that PGC overexpression can downregulate IQGAP1 protein expression and significantly inhibit migration of GC cells, and that effect can be restored by IQGAP1 overexpression, which suggested that the regulation of IQGAP1 by PGC plays an important role in inhibiting the migration and invasion of GC cells. As a key node of the small GTPase network, IQGAP1 is a major regulator of several GTPases, of which the Rho family is the most important class of small GTPases that interact with IQGAP1 [31]. Multiple studies have proven that IQGAP1 is significantly associated with the Rho-GTPase pathway, and the binding and regulation of its GRD domain to the Rho-GTPase pathway proteins plays an important role in tumor invasion and metastasis [32,33,34,35]. In this study, the relationship between PGC-IQGAP1 and the Rho-GTPase pathway was investigated. Compared to the control group, PGC overexpression downregulated not only the IQGAP1 protein but also the Rho protein expression. Based on the above research results, we hypothesize that one of the mechanisms by which PGC inhibits the migration and invasion of GC cells may be by promoting the downregulation of the IQGAP1 protein via interacting with IQGAP1 and thereby inhibiting the expression of Rho protein in the Rho-GTPase pathway, with ultimate inhibitory effects on GC cell migration and invasion. In addition, we found that the upregulation of PGC can shorten the half-life of the IQGAP1 protein, suggesting that PGC can reduce the stability of the IQGAP1 protein. Studies have found that the high ubiquitination level of IQGAP1 can reduce its expression and its interaction with and activation of Rho-GTPase, thereby inhibiting tumor metastasis. Further experiments are needed to verify whether PGC inhibits GC cell migration and invasion by affecting the ubiquitination level of IQGAP1.

## 5. Conclusions

PGC is negatively correlated with malignant biological behaviors of GC cells, such as proliferation, viability, stemness, migration, invasion, EMT, tumorigenesis, and intraperitoneal dissemination, but it is positively correlated with the maturation and differentiation of GC cells. PGC may act as a tumor suppressor gene in the development, invasion, and metastasis of GC. PGC can downregulate its interacting protein IQGAP1 and inhibit the Rho-GTPase pathway, thereby participating in the inhibition of GC cell migration and invasion. PGC is expected to become a new therapeutic target, further exerting its role beyond being an early warning marker of GC and its precancerous diseases (Figure 7).

## Figures and Tables

**Figure 1 cancers-16-01796-f001:**
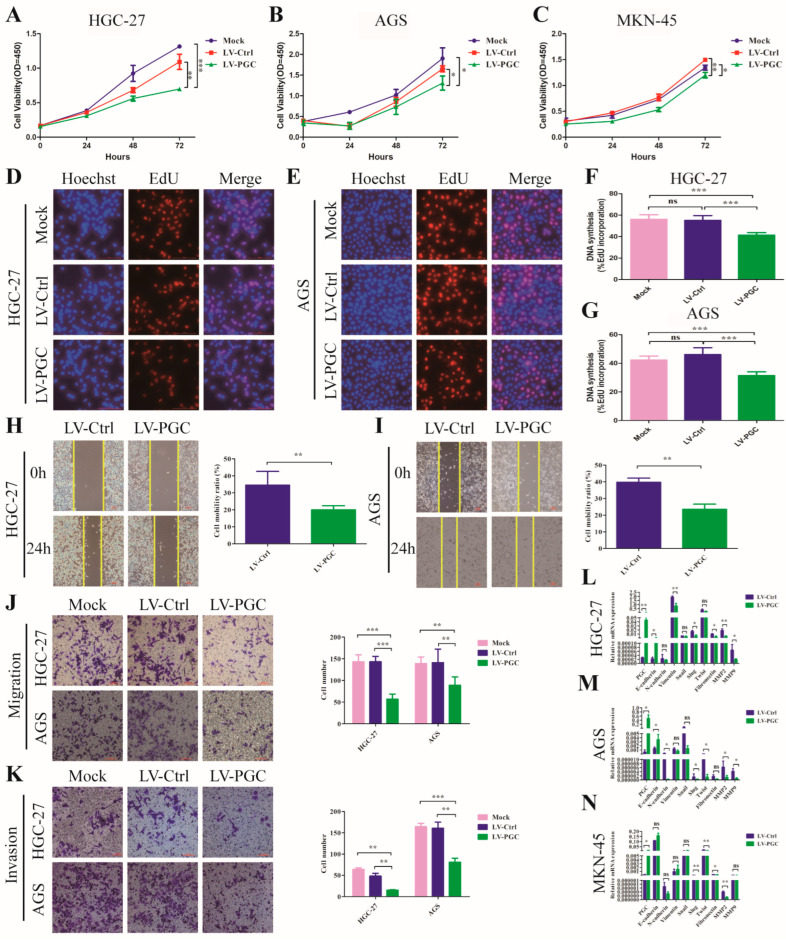
PGC inhibits GC cell proliferation, migration, invasion, and EMT in vitro. (**A**–**C**) Cell viability was measured by using CCK-8 assay after the transfection of HGC-27 (**A**), AGS (**B**), and MKN-45 (**C**) cells with LV-PGC and LV-Ctrl, and the results suggested that PGC inhibited cell viability. (**D**–**G**) Cell proliferation was evaluated by using EdU assay after the transfection of HGC-27 and AGS cells with LV-PGC and LV-Ctrl. Representative images of EdU assays (**D**,**E**) and the EdU incorporation percentage (**F**,**G**) revealed that overexpression of PGC inhibited cell proliferation. (**H**,**I**) Cell motility was examined with a wound healing assay in HGC-27 and AGS cells transfected with LV-PGC and LV-Ctrl. Representative images and the cell mobility ratio revealed that PGC inhibited GC cell migration. (**J**,**K**) Migration and invasion of HGC-27 and AGS cells transfected with LV-PGC and LV-Ctrl were determined using transwell migration and invasion assays. Representative images of transwell assays and the number of cells passing through the chamber revealed that PGC inhibited GC cell migration and invasion. (**L**–**N**) qRT-PCR analysis of the expression levels of EMT-associated markers E-cadherin, N-cadherin, vimentin, Snail, Slug, Twist, fibronectin, MMP2, and MMP9 in HGC-27 (**L**), AGS (**M**), and MKN-45 (**N**) cells transfected with LV-PGC and LV-Ctrl. The results suggested that PGC regulated the expression of EMT biomarkers. Data are shown as means ± SEM; * *p* < 0.05, ** *p* < 0.01, *** *p* < 0.001, ns: no significance, scale bar, 100 mm.

**Figure 2 cancers-16-01796-f002:**
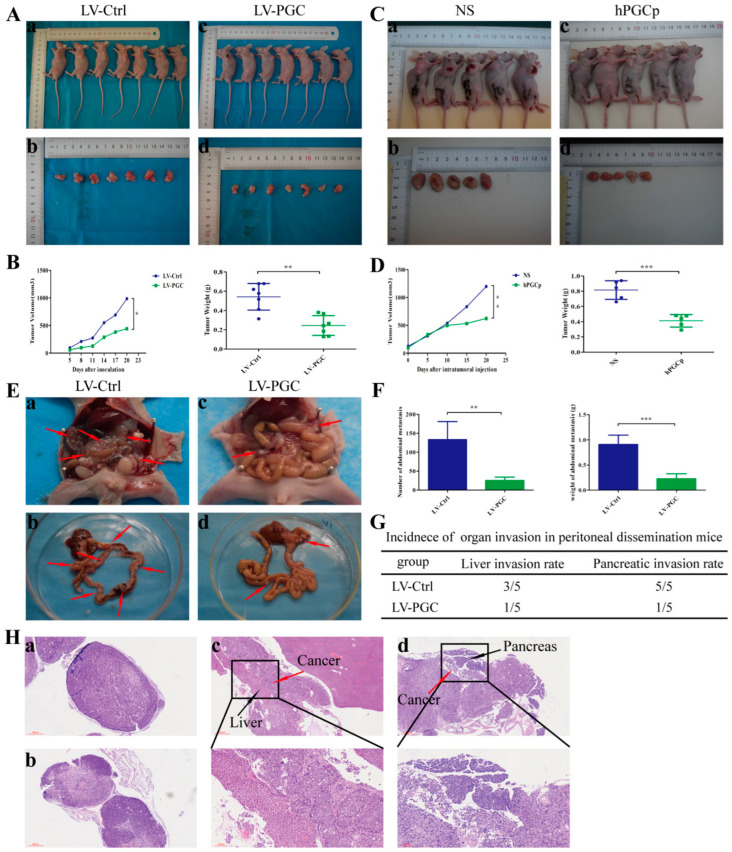
PGC inhibits GC cell growth and metastasis in vivo. (**A**,**B**) MKN-45 cells stably expressing PGC (LV-PGC) showed inhibited subcutaneous tumor growth compared with the negative control vector group (LV-Ctrl). Representative images of the macroscopic appearance of subcutaneous nodules (**A**) and the tumor volume over the entire period and tumor weight at the end point (**B**) (n = 7 per group). (**C**,**D**) MKN-45 cells were subcutaneously injected into nude mice to create a subcutaneous tumor model. Intratumoral injection of hPGCp significantly inhibited subcutaneous tumor growth compared to the injection of the NS group. Representative images of the macroscopic appearance of subcutaneous nodules (**C**) and the tumor volume over the entire period and tumor weight at the endpoint (**D**) (n = 5 per group). (**E**–**H**) MKN-45 cells stably expressing PGC (LV-PGC) showed inhibited peritoneal dissemination compared to the negative control vector group (LV-Ctrl). Representative images of the macroscopic appearance of peritoneal dissemination nodules, red arrows indicate the representative peritoneal dissemination nodules. (**E**), the number and weight of abdominal metastasis nodules (**F**), the incidence of organ invasion in mice with peritoneal dissemination mice (**G**), and H&E staining of peritoneal dissemination nodules as well as the invasion of the liver and pancreas (**H**). (**a**,**b**) show pathological images of abdominal metastasis nodules in the LV-Ctrl and LV-PGC groups, respectively, while (**c**,**d**) show representative pathological images of liver invasion and pancreatic invasion in the LV-Ctrl group (n = 5 per group). Data are shown as means ± SEM; * *p* < 0.05, ** *p* < 0.01, *** *p* < 0.001.

**Figure 3 cancers-16-01796-f003:**
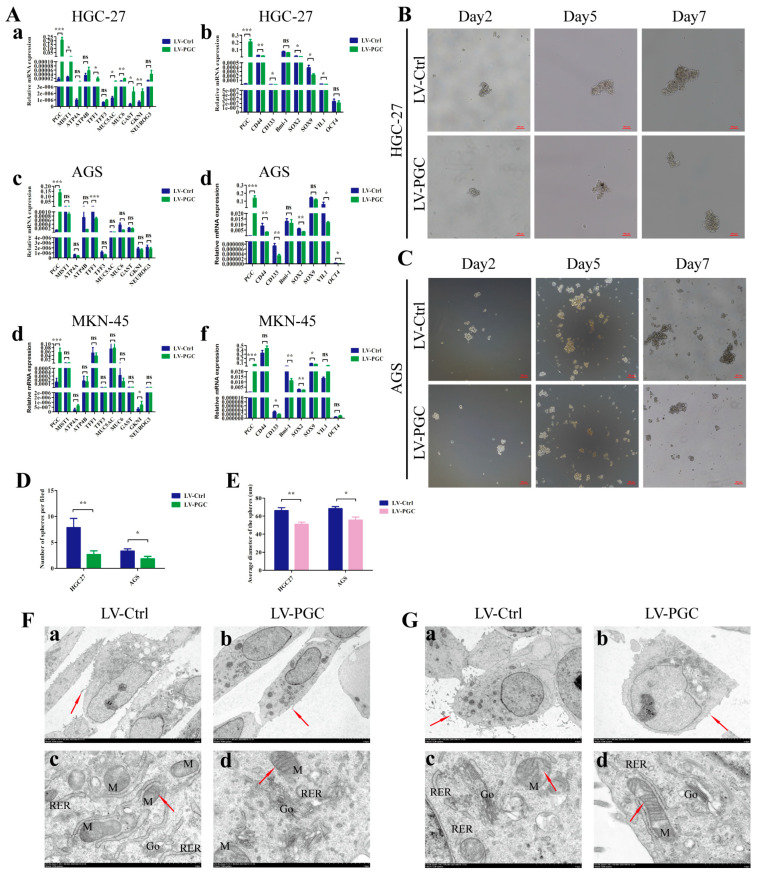
PGC promotes differentiation and inhibits the stemness of GC cells. (**A**) qRT-PCR analysis of the expression levels of mature differentiation markers and dedifferentiation markers in gastric mucosa epithelium in HGC-27 (**a**,**b**), AGS (**c**,**d**), and MKN-45 (**e**,**f**) cells transfected with LV-PGC and LV-Ctrl. The results revealed that PGC promoted the expression of mature differentiation markers of gastric mucosa but inhibited the expression of biomarkers of progenitor cells and GC stem cells. (**B**–**E**) The stem-cell-like phenotype of HGC-27 and AGS cells transfected with LV-PGC and LV-Ctrl was determined using sphere formation assays. Representative images of spheres on day 2, day 5, and day 7 (**B**,**C**), the number of spheres per field (**D**), and the average sphere diameter (**E**) showed that PGC significantly reduced the sphere-forming efficiency of GC cells. (**F**,**G**) Representative images of the subcellular morphological structure characteristics of HGC-27 (**F**) and AGS (**G**) cells revealed that PGC-overexpressing cells exhibited more differentiated subcellular structure characteristics. (**a**,**b**) Microvilli on the cell surface (shown by the red arrows, magnification, ×2.0 k). (**c**,**d**) The intracellular mitochondria (M) (shown by the red arrows) and mitochondrial crista, rough endoplasmic reticulum (RER), and Golgi apparatus (Go) (magnification, ×15.0 k). Data are shown as means ± SEM; * *p* < 0.05, ** *p* < 0.01, *** *p* < 0.001, ns: no significance.

**Figure 4 cancers-16-01796-f004:**
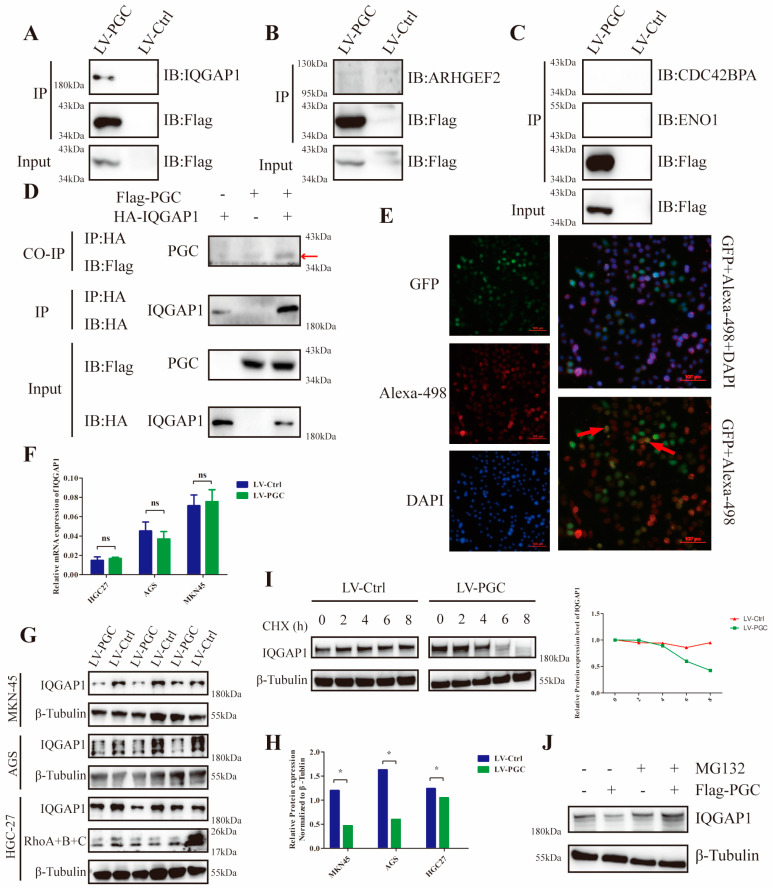
PGC interacts with IQGAP1 and interferes with IQGAP1 protein stability in GC cells. For the Co-IP assays, AGS cells were transfected with FLAG-PGC, and the FLAG fusion magnetic beads were used for pulldown analysis. (**A**–**C**) FLAG-PGC interacts with IQGAP1 (**A**), but not ARHGEF2 (**B**), CDC42BP2, or ENO1 (**C**). (**D**) HA-IQGAP1 interacts with FLAG-PGC in GC cells, indicating that PGC and IQGAP1 can interact with each other. The AGS cells were cotransfected with HA-IQGAP1 and FLAG-PGC. HA fusion magnetic beads were used for pull-down analysis, and anti-HA antibodies as well as anti-FLAG antibodies were used for WB. The red arrow points to the band of interest. (**E**) PGC and IQGAP1 are co-localized in the cytoplasm, as demonstrated by immunofluorescence staining analysis. The red arrows indicate the representative overlap of green fluorescence with red fluorescence. (**F**–**H**) qRT-PCR and WB were used to determine the expression level changes of IQGAP1 in HGC-27, AGS, and MKN-45 cells transfected with LV-PGC and LV-Ctrl. The results showed that PGC decreased the expression of the IQGAP1 protein rather than the mRNA, which suggested that the effect of PGC on IQGAP1 may be at the post-transcriptional level. (**I**) AGS cells treated with CHX (100 µM/mL) revealed that PGC overexpression accelerated the degradation of IQGAP1. (**J**) AGS cells treated with MG132 for 6 h showed that the downregulation of IQGAP1 protein induced by PGC was rescued by the proteasome inhibitor MG132. PGC may decrease the protein stability of IQGAP1 by affecting its ubiquitination level. Data are shown as means ± SEM; * *p* < 0.05, ns: no significance. The uncropped bolts are shown in Appendix A.

**Figure 5 cancers-16-01796-f005:**
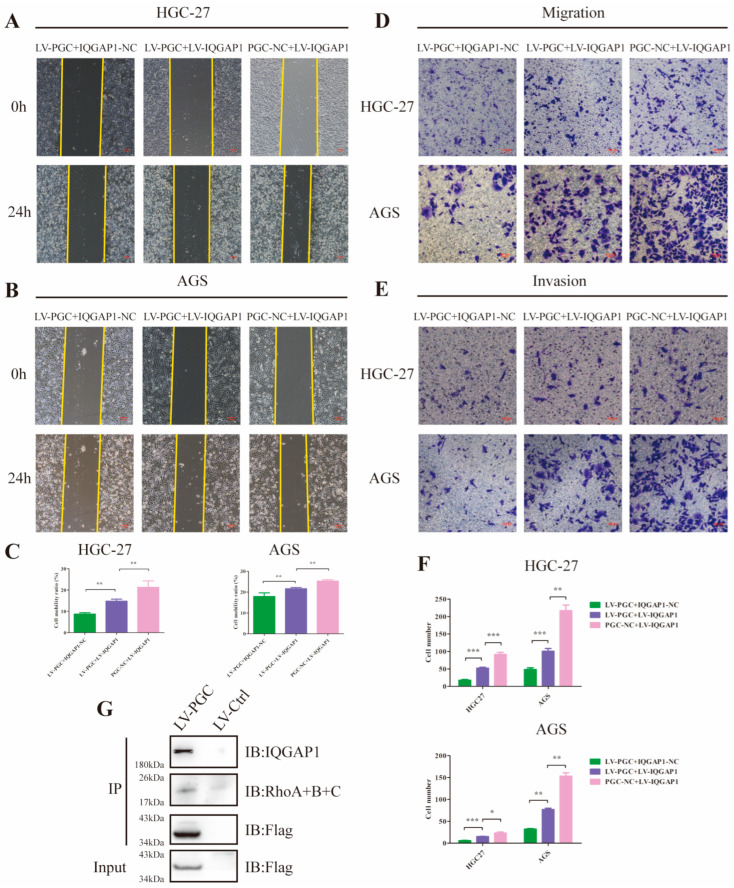
PGC inhibits GC cell migration and invasion by reducing IQGAP1 and suppressing Rho-GTPase signaling. (**A**–**C**) Cell motility was examined using wound healing assays in HGC-27 and AGS cells transfected with LV-PGC + IQGAP1-NC, PGC-NC + LV-IQGAP1, and LV-PGC + LV-IQGAP1. Representative images (**A**,**B**) and the cell mobility ratio (**C**) revealed that increased expression of IQGAP1 counteracted the effects of PGC overexpression on cell motility. (**D**–**F**) The migration and invasion of HGC-27 and AGS cells transfected with LV-PGC + IQGAP1-NC, PGC-NC + LV-IQGAP1, and LV-PGC + LV-IQGAP1 were determined using transwell migration and invasion assays. Representative images (**D**,**E**) of transwell assays and the numbers of cells passing through the chamber (**F**) revealed that IQGAP1 counteracted the effects of PGC overexpression on cell migration and invasion. Data are shown as means ± SEM; * *p* < 0.05, ** *p* < 0.01, *** *p* < 0.001. (**G**) FLAG-PGC can interact with Rho. AGS cells were transfected with FLAG-PGC. For the Co-IP assays, FLAG fusion magnetic beads were used for pull-down analysis, and anti-RhoA + B + C antibody was used for WB. The uncropped bolts are shown in Appendix A.

**Figure 6 cancers-16-01796-f006:**
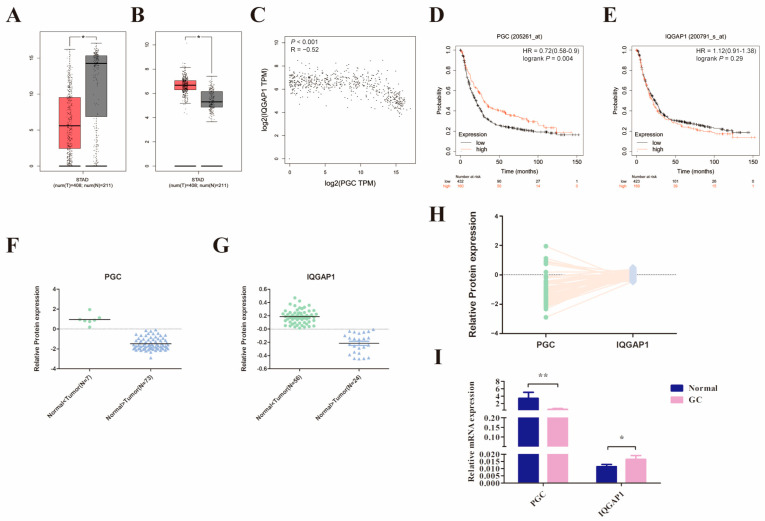
PGC expression in GC is significantly negatively correlated with IQGAP1. (**A**–**C**) The mRNA expression level of *PGC* (**A**) and *IQGAP1* (**B**) and the correlation of their expression (**C**) were determined by using the GEPIA2 database, which suggested that *PGC* expression is negatively correlated with *IQGAP1* at the mRNA level. The red section represents tumor tissue, and the gray section represents normal tissue. The data are shown as means ± SEM; * *p* < 0.05. (**D**,**E**) The prognostic roles of *PGC* (**D**) and *IQGAP1* (**E**) mRNA expression in GC were determined by using the Kaplan-Meier Plotter database. (**F**–**H**) The protein expression levels of PGC (**F**) and IQGAP1 (**G**) and the expression correlation (**H**) were determined by using the CPTAC database, which showed that PGC expression is negatively correlated with IQGAP1 at the protein level. (**I**) The mRNA expression data of 20 paired GC tissues and the correlated normal tissues revealed that *PGC* was downregulated while *IQGAP1* was upregulated in GC tissues. The data are shown as means ± SEM; * *p* < 0.05, ** *p* < 0.01.

**Figure 7 cancers-16-01796-f007:**
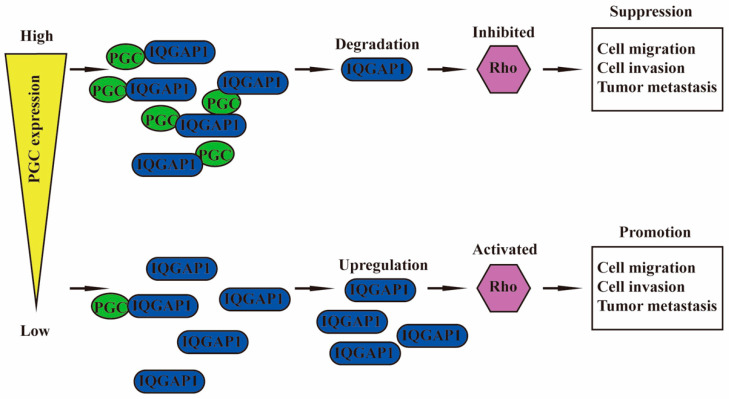
Schematic diagram summarizing the function of PGC on GC. PGC can downregulate its interacting protein IQGAP1 and inhibit the Rho-GTPase pathway, thereby participating in the inhibition of GC cell migration and invasion.

## Data Availability

The original contributions presented in the study are included in the article/Appendix A. Further inquiries can be directed to the corresponding authors.

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
