# Peer review of "Pepsinogen C Interacts with IQGAP1 to Inhibit the Metastasis of Gastric Cancer Cells by Suppressing Rho-GTPase Pathway"

_cancers, 2024, doi:10.3390/cancers16101796_

Round 1
Reviewer 1 Report
Comments and Suggestions for Authors
This manuscript has investigated PGC and one of its protein partners IQGAP1 in gastric cancer cells. In this study the authors have employed a range of in vitro and in vivo techniques to provide evidence supporting an inhibitory function of PGC in gastric cancer cells that may in part be overcome by IQGAP1 overexpression. The types of experiments included in the study are appropriate for this type of work and appear to have been performed according to established guidelines. Figures are well-presented, the text and referencing are good and the findings are interesting and worthy of further study. I recommend this article for publication pending amendments as detailed below. These are not major issues.
1. In Figure 1, the authors have shown results for all 3 gastric cancer cell lines included in this study (HGC-27, AGS, MKN-45) in CCK-8 assays and EMT marker RNA expression, but only HGC-27 and AGS for EdU, wound healing and invasion/migration assays. As the xenograft model that the authors subsequently use is based on the MKN-45 cell line, results from this cell line should also be shown. The authors should also briefly describe why they chose the MKN-45 cell line for xenograft studies, especially as these cells seemed to show the least effects of PGC over-expression (Figure 1C).
2. Throughout the manuscript, the authors have depicted a different cell type in each set of experiments and it is not clear whether all experiments have been performed on all of the 3 cell types with the best results depicted, or whether results were variable between the cell types with the results that best fit the direction of the manuscript included, or whether the authors used different cell types for each set of experiments. Use of the 3 cell lines in each of the in vitro experiments, the consistency (or lack of consistency) of results between the cell lines and results of these experiments should be added to the text, with results images included as part of the main text/Figures or as Supplementary figures. This would indicate that results may be broadly applicable to gastric cancer cells and are not cell line specific.
3. It is noticeable that the labelling of blots in the Supplementary Figures is better than that in the manuscript, in particular the molecular sizes of bands. The uppermost blot image in Figure 4D is not clear and requires additional labelling to indicate the band of interest (i.e. an arrow). The band should be centred on the blot image.
4. The CCK-8 assay is an indirect way of measuring viable cell number in a culture. Because cell numbers (depicted in Figure 1A) were increasing at every timepoint, it does not appear that PGC overexpression was affecting cell viability, rather that the rate of cell growth was reduced, a finding reflected by the reduced DNA synthesis evident in EdU assays. In the Supplementary data, the authors show that the reduced growth of PGC overexpressing cells was not associated with accumulation of cells in a particular phase of the cell cycle. (This suggests that either reduced numbers of cells were entering the cell cycle or that the cell cycle overall was slowed. Different types of experiments would be required to characterise cell cycle effects, but this is not a focus of the present study).
5. What was PGC expression in the gastric cancer xenografts? (Was PGC expression maintained in the xenograft tissues)?
6. In the Discussion section, the sentence in lines 421 – 423 seems to be an overstatement as PGC testing has not proceeded to clinical use in the 15 years since the first publication. Although it remains an area of interest, the 3 studies that have included PGC do not constitute “numerous clinical studies”. It is suggested that the sentence is modified to better reflect the available data.
7. Almost all of the Discussion section is a summary of the results obtained by the authors. The sentence in lines 468-470 is misleading as it implies that the 9 proteins listed interact with PGC when only IQGAP was shown to be a bona fide interactor, both ARHGEF2 and CDC42BPA did not co-immunoprecipitate with PGC (Figure 4B,C) and no evidence was presented to support interaction of PGC with the remaining proteins. This and the following sentence need to be changed to accurately reflect the results obtained.
8. There is an error in line 499 as there is no reference 53.
Comments on the Quality of English LanguageThe manuscript requires proper English language editing. Apart from grammatical errors throughout the text, a number of sentences are incomplete or do not make sense.
Author Response
Dear reviewer,
Thank you very much for your valuable and precious suggestions and reminding. We answered the suggestions point by point as follows and ticked the revision in the purple color.
Yours sincerely,
Yuan Yuan
Comments and Suggestions for Authors
This manuscript has investigated PGC and one of its protein partners IQGAP1 in gastric cancer cells. In this study the authors have employed a range of in vitro and in vivo techniques to provide evidence supporting an inhibitory function of PGC in gastric cancer cells that may in part be overcome by IQGAP1 overexpression. The types of experiments included in the study are appropriate for this type of work and appear to have been performed according to established guidelines. Figures are well-presented, the text and referencing are good and the findings are interesting and worthy of further study. I recommend this article for publication pending amendments as detailed below. These are not major issues.
- In Figure 1, the authors have shown results for all 3 gastric cancer cell lines included in this study (HGC-27, AGS, MKN-45) in CCK-8 assays and EMT marker RNA expression, but only HGC-27 and AGS for EdU, wound healing and invasion/migration assays. As the xenograft model that the authors subsequently use is based on the MKN-45 cell line, results from this cell line should also be shown. The authors should also briefly describe why they chose the MKN-45 cell line for xenograft studies, especially as these cells seemed to show the least effects of PGC over-expression (Figure 1C).
Response 1: Thank you for your question. In this study, we selected a total of three different gastric cancer(GC)cell lines as research objects, including moderately differentiated GC cell line AGS and undifferentiated GC cell line HGC27, both of which are adherent cells. There is also a poorly differentiated GC cell line MKN45, which is semi-suspended cells. The reason for selecting three GC cell lines is to explore whether the experimental results are cell generalizing or cell line specific. It should be noted that not all experiments used all three GC cell lines. In EdU, wound healing, and invasion/migration that required the detection of adherents, we used AGS and HGC27, while in CCK8 and qPCR that were not affected by suspension properties, we used all three cell lines. Considering that MKN-45's semi-suspended growth characteristics may be limited by cell colonization and adhesion, which may affect the accuracy of experimental results, we did not use MKN-45 for EdU, wound healing and invasion/migration experiments. In turn, we use MKN-45 for xenotransplantation experiment because the MKN-45 is more invasive and more prone to tumor formation in animal models. Among the three cell lines, although the overexpression of PGC in MKN45 cells is the smallest, its overexpression can reach more than 100 times, which is stable and reliable for transplantation tumor experiments.
- Throughout the manuscript, the authors have depicted a different cell type in each set of experiments and it is not clear whether all experiments have been performed on all of the 3 cell types with the best results depicted, or whether results were variable between the cell types with the results that best fit the direction of the manuscript included, or whether the authors used different cell types for each set of experiments. Use of the 3 cell lines in each of the in vitro experiments, the consistency (or lack of consistency) of results between the cell lines and results of these experiments should be added to the text, with results images included as part of the main text/Figures or as Supplementary figures. This would indicate that results may be broadly applicable to gastric cancer cells and are not cell line specific.
Response 2: Thank you for your professional advice. As mentioned above, in our study, we chose different cell lines for in vitro experiments. Due to the semi-suspension characteristics of the MKN45 cell line, we only used the adherent cell lines HGC27 and AGS in experiments such as EdU, wound healing, invasion/migration assays, and sphere formation. However, in experiments such as CCK8 and qPCR, which are not affected by suspension characteristics, we used all three cell lines, HGC27, AGS, and MKN45. To avoid confusion, we detailed in the revised manuscript which cell line was used for each in vitro experiment in Section 3.1, 3.3, 3.4 and 3.6. It is worth noting that the results of each in vitro experiment are derived from at least HGC27 and AGS cell lines, indicating that our research results are broadly applicable to gastric cancer cells, not limited to specific cell lines.
- It is noticeable that the labelling of blots in the Supplementary Figures is better than that in the manuscript, in particular the molecular sizes of bands. The uppermost blot image in Figure 4D is not clear and requires additional labelling to indicate the band of interest (i.e. an arrow). The band should be centred on the blot image.
Response 3: Thank you for your reminder. In the revised version of the manuscript, we have modified Figure 4 in page 12 and Figure 5 in page 14, labeled the molecular sizes of the bands, and added arrows to indicate the bands of interest.
- The CCK-8 assay is an indirect way of measuring viable cell number in a culture. Because cell numbers (depicted in Figure 1A) were increasing at every timepoint, it does not appear that PGC overexpression was affecting cell viability, rather that the rate of cell growth was reduced, a finding reflected by the reduced DNA synthesis evident in EdU assays. In the Supplementary data, the authors show that the reduced growth of PGC overexpressing cells was not associated with accumulation of cells in a particular phase of the cell cycle. (This suggests that either reduced numbers of cells were entering the cell cycle or that the cell cycle overall was slowed. Different types of experiments would be required to characterise cell cycle effects, but this is not a focus of the present study).
Response 4: Thank you for your valuable suggestions. Your observations regarding the CCK-8 and EdU experiments indeed shed light on the impact of PGC overexpression on cell growth rate and the significance of studying cell cycle effects. Supplementary data indicate that the decelerated growth of PGC-overexpressing cells is not accompanied by accumulation in specific cell cycle phases, suggesting that the reduced growth may stem from a decrease in the number of cells entering the cell cycle or a slowdown in the overall cell cycle. Although this study did not delve deeply into cell cycle effects, your suggestions provide valuable directions for future research. We have also added relevant discussions at lines 445-448 in the revised manuscript.
- What was PGC expression in the gastric cancer xenografts? (Was PGC expression maintained in the xenograft tissues)?
Response 5: Thank you for your question. According to your suggestion, we have supplemented immunohistochemistry images of PGC expression in the gastric cancer xenografts (Figure S2), demonstrating that PGC expression is still maintained in the xenograft tissues. The relevant description has been added to lines 233-234 of the revised manuscript.
- In the Discussion section, the sentence in lines 421 – 423 seems to be an overstatement as PGC testing has not proceeded to clinical use in the 15 years since the first publication. Although it remains an area of interest, the 3 studies that have included PGC do not constitute “numerous clinical studies”. It is suggested that the sentence is modified to better reflect the available data.
Response 6: Thank you for the reminder. We have revised the relevant sentences as follows: ‘Several investigations have validated the potential of serum PGC expression levels as indicators for screening GC and its precancerous conditions, holding significant promise for clinical applications’. The relevant content in the manuscript has also been modified, as detailed in lines 428-430.
- Almost all of the Discussion section is a summary of the results obtained by the authors. The sentence in lines 468-470 is misleading as it implies that the 9 proteins listed interact with PGC when only IQGAP was shown to be a bona fide interactor, both ARHGEF2 and CDC42BPA did not co-immunoprecipitate with PGC (Figure 4B,C) and no evidence was presented to support interaction of PGC with the remaining proteins. This and the following sentence need to be changed to accurately reflect the results obtained.
Response 7: Thank you for the reminder. We have revised the relevant sentences as follows: ‘Among these 32 proteins, we identified 9 proteins that may overlap with PGC in sub-cellular localization, including ARHGEF2, CDC42BPA, ENO1, IQGAP1, EPHA2, GPI, MYH9, S100A14, and YBX1. Among them, ARHGEF2, CDC42BP2, ENO1, and IQGAP1 are closely associated with the Rho-GTPase network and play pivotal roles in cytoskeletal organization. Furthermore, through two-way Co-IP and immunofluorescence experiments, we found that PGC interacts only with IQGAP1 among these four proteins and co-localizes with it.’ The relevant content in the manuscript has also been modified, as detailed in lines 476-482.
- There is an error in line 499 as there is no reference 53.
Response 8: Thank you for your reminding. We have removed reference 53 and reviewed all the references to ensure accuracy.
Reviewer 2 Report
Comments and Suggestions for Authors
The article entitled “Pepsinogen C Interacts with IQGAP1 to Inhibit the Metastasis of Gastric Cancer Cells by Suppressing Rho-GTPase pathway” is an interesting article on the elucidation of the possible new role of PGC as a tumor suppressor.
The authors have explored an additional possible role of PGC in this article where they show it as a tumor suppressor. They have used 3 different GC cell lines to study proliferation, migration, invasion, and EMT in vitro and studied GC cell growth and metastasis in vivo to demonstrate the tumor suppressive effect of PGC on GC. They also show the role of PGC in promoting differentiation and inhibiting the stemness of GC cells. Further, the mechanistic insight showed the interaction and degradation of PGC with IQGAP1 protein through suppression of the Rho-GTPase signaling pathway.
The study is well-planned and executed, supported by sufficient experimental data. More in vivo experiments would have increased the significance of the findings. Nevertheless, the presented data is enough for the current manuscript.
The authors need to address the following comments:
1. There are a lot of grammatical errors that need to be rectified. For example, in lines 24-25 “Thus, PGC was potential to be a new therapeutic target for gastric cancer.”
2. Multiple sentences convey a wrong/unclear message. For example, in Lines 36-37, “In addition, PGC shortened the half-life of IQGAP1 protein to downregulate its expression level.” In this sentence, it's not clear whether the protein is degraded before completing its lifetime, or the expression of the protein is downregulated.
3. The manuscript is scientifically not presented well and needs a better presentation.
4. Why have the authors chosen 3 GC cell lines? What’s the rationale for choosing them? How are they different in terms of deducing the scientific outcome? Authors should write briefly about the choice of their experimental models.
5. Why are the GC cells' DNA incorporation, migration, and invasion checked only in two cell lines and not the third?
6. Why was the in vivo experiment as in Figure 2 carried out using only MKN-45 cells and not the other 2 cell lines?
7. Authors should explain what hPGCp is at its first mention.
8. The qRT-PCR results of the differentiation and stemness markers are not consistent in all the 3 cell lines. For example, the MIST1 expression is significant in HGC-27 but not in AGS and MKN-45. Likewise, many other gene expressions for both the markers. How do the authors explain this variability with the different cell lines?
Comments on the Quality of English Language
Extensive editing is required.
Author Response
Dear reviewer,
Thank you very much for your valuable and precious suggestions and reminding. We answered the suggestions point by point as follows and ticked the revision in the purple color.
Yours sincerely,
Yuan Yuan
Comments and Suggestions for Authors
The article entitled “Pepsinogen C Interacts with IQGAP1 to Inhibit the Metastasis of Gastric Cancer Cells by Suppressing Rho-GTPase pathway” is an interesting article on the elucidation of the possible new role of PGC as a tumor suppressor.
The authors have explored an additional possible role of PGC in this article where they show it as a tumor suppressor. They have used 3 different GC cell lines to study proliferation, migration, invasion, and EMT in vitro and studied GC cell growth and metastasis in vivo to demonstrate the tumor suppressive effect of PGC on GC. They also show the role of PGC in promoting differentiation and inhibiting the stemness of GC cells. Further, the mechanistic insight showed the interaction and degradation of PGC with IQGAP1 protein through suppression of the Rho-GTPase signaling pathway.
The study is well-planned and executed, supported by sufficient experimental data. More in vivo experiments would have increased the significance of the findings. Nevertheless, the presented data is enough for the current manuscript.
The authors need to address the following comments:
- There are a lot of grammatical errors that need to be rectified. For example, in lines 24-25 “Thus, PGC was potential to be a new therapeutic target for gastric cancer.”
Response 1: Thank you for the reminder. We have revised the sentence in lines 26-27 to: "Thus, PGC has the potential to be a new therapeutic target for gastric cancer."
- Multiple sentences convey a wrong/unclear message. For example, in Lines 36-37, “In addition, PGC shortened the half-life of IQGAP1 protein to downregulate its expression level.” In this sentence, it's not clear whether the protein is degraded before completing its lifetime, or the expression of the protein is downregulated.
Response 2: Thank you for your reminding. We have revised the sentence in lines 38-39 to: "PGC disrupts the stability of the IQGAP1 protein, promoting its degradation and significantly shortening its half-life."
- The manuscript is scientifically not presented well and needs a better presentation.
Response 3: Thank you for your professional comment and constructive suggestion. As we mentioned in the introduction, PGC has long been regarded as a final product of the mature differentiation of gastric mucosal cells, and few studies have explored its impact as a regulatory factor on the biological behavior of cancer cells and its molecular mechanisms. A comprehensive and in-depth exploration and evaluation of the effects of PGC on cancer cells’ biological functions and the underlying molecular mechanisms will help to explain the role of PGC in the occurrence and development of cancer, and provide a new theoretical basis for the prevention, diagnosis and treatment of cancer and also deliver a potential therapeutic target. In this study, we systematically investigated the effect of PGC on the biological functions of GC cells for the first time, both in vivo and in vitro. At the same time, we attempted to identify proteins that may interact with PGC in GC cells and whether their interactions inhibit the migration and invasion of GC cells. Our goal is to clarify whether PGC plays a regulatory role in GC, and whether, in addition to being an early warning marker for GC and its precancerous diseases, PGC has the potential to become a new therapeutic target for GC. To the best of our knowledge, this is the first study to report the biological function of PGC beyond its original role as a precursor to digestive enzymes. According to your suggestions, we have appropriately rewritten the introduction and discussion in the revised version to further highlight the scientific significance of this study.
- Why have the authors chosen 3 GC cell lines? What’s the rationale for choosing them? How are they different in terms of deducing the scientific outcome? Authors should write briefly about the choice of their experimental models.
Response 4: Thank you for your question and suggestion. We chose three gastric cancer cell lines to demonstrate the broad applicability of our experimental results to gastric cancer cells, rather than specific to any particular cell line. Among these three cell lines, HGC27 cells represent undifferentiated gastric cancer cells and are adherent, AGS cells represent moderately differentiated gastric cancer cells and are adherent, while MKN45 represents poorly differentiated gastric cancer cells and grows in semi-suspension. In experiments such as EdU, wound healing, and invasion/migration assays, where adherent cells are required, we used AGS and HGC27 cell lines. However, in experiments such as CCK8 and qPCR, which are not influenced by suspension characteristics, we used all three cell lines. To avoid confusion, we have provided additional explanations in the revised manuscript regarding the cell lines used in each experiment.
- Why are the GC cells' DNA incorporation, migration, and invasion checked only in two cell lines and not the third?
Response 5: Thank you for your question. We did not use the MKN45 cell line for DNA incorporation, migration, and invasion assays because we worry about its unique semi-suspension characteristics may be limited by cell colonization and adhesion, which could potentially affect the reliability of the experimental results. Therefore, we conducted these assays in the adherent HGC27 and AGS cell lines.
- Why was the in vivo experiment as in Figure 2 carried out using only MKN-45 cells and not the other 2 cell lines?
Response 6: Thank you for your question. The reason for using MKN-45 cells instead of the other two cell lines for in vivo experiments is because MKN-45 cells exhibit stronger invasive capabilities and are more likely to form tumors in animal models.
- Authors should explain what hPGCp is at its first mention.
Response 7: Thank you for the reminder. "hPGCp" refers to human PGC active protein, and its full name has been added to the revised manuscript at line 238.
- The qRT-PCR results of the differentiation and stemness markers are not consistent in all the 3 cell lines. For example, the MIST1 expression is significant in HGC-27 but not in AGS and MKN-45. Likewise, many other gene expressions for both the markers. How do the authors explain this variability with the different cell lines?
Response 8: Thank you for your question. As you mentioned, there is indeed inconsistency in the regulation of PGC on the expression of differentiation markers in the three gastric cancer cell lines, which we conjecture that is mainly due to their respective stages of differentiation. Specifically, HGC27 is an undifferentiated gastric cancer cell line, MKN45 is a low-differentiated gastric cancer cell line. and AGS is a moderately differentiated gastric cancer cell line. This difference in differentiation stages likely accounts for the inconsistency in the expression of differentiation markers. Despite these limitations, our results indicate that PGC predominantly upregulates the expression of differentiation markers in GC cells. Combined with the observation that HGC-27 and AGS cells overexpressing PGC exhibit subcellular structural features indicative of a higher degree of differentiation, as shown in Figure 3F-G, we can conclude that PGC promotes GC cell differentiation. On the other hand, regarding stemness markers, our results demonstrate that in most cases, PGC downregulates the expression of stemness markers in GC cells. Specifically, CD133 and SOX2 are significantly downregulated in all three GC cell lines overexpressing PGC, while CD44, SOX9, and VIL1 are significantly downregulated in at least two cell lines, and Bmi1 and OCT4 are significantly downregulated in one cell line each. Combined with the observation that PGC inhibits the formation of GC cell spheroids, as shown in Figure 3B-E, we believe that PGC suppresses the stemness of GC cells. In summary, we appreciate your valuable feedback once again. Abnormalities in GC cell differentiation and stemness often involve complex regulatory mechanisms that are closely related to tumor occurrence, progression, and prognosis. Therefore, further investigation into the underlying mechanisms of how PGC affects GC cell differentiation and stemness represents an important research direction for the future. In the revised version results section 3.3, we have added Supplementary Table 2. The expression of differentiation and stemness markers in gastric cancer cells with PGC overexpression.
Round 2
Reviewer 1 Report
Comments and Suggestions for Authors
I feel that the authors have appropriately addressed reviewers' questions and comments, and that the manuscript is suitable for publication.
Comments on the Quality of English LanguageEnglish language editing will be required to correct minor grammatical errors.
Reviewer 2 Report
Comments and Suggestions for Authors
The authors have addressed the raised questions in a clear and detailed manner.